# Data Sampling Affects the Complexity of Online SGD over Dependent Data

**Shaocong Ma**[1]    **Ziyi Chen**[1]    **Yi Zhou**[1]    **Kaiyi Ji**[2]    **Yingbin Liang**[3]

[1]Department of Electrical and Computer Engineering, University of Utah
[2]Electrical Engineering and Computer Science Department, University of Michigan, Ann Arbor
[3]Department of Electrical and Computer Engineering, The Ohio State University

## Abstract

Conventional machine learning applications typically assume that data samples are independently and identically distributed (i.i.d.). However, practical scenarios often involve a data-generating process that produces highly dependent data samples, which are known to heavily bias the stochastic optimization process and slow down the convergence of learning. In this paper, we conduct a fundamental study on how different stochastic data sampling schemes affect the sample complexity of online stochastic gradient descent (SGD) over highly dependent data. Specifically, with a $\phi$-mixing process of data, we show that online SGD with proper periodic data-subsampling achieves an improved sample complexity over the standard online SGD in the full spectrum of the data dependence level. Interestingly, even subsampling a subset of data samples can accelerate the convergence of online SGD over highly dependent data. Moreover, we show that online SGD with mini-batch sampling can further substantially improve the sample complexity over online SGD with periodic data-subsampling over highly dependent data. Numerical experiments validate our theoretical results.

## 1 INTRODUCTION

Stochastic optimization algorithms have attracted great attention in the past decade due to its successful applications to a broad research areas, including deep learning [Goodfellow et al., 2016], reinforcement learning [Sutton and Barto, 2018], online learning [Bottou, 2010, Hazan, 2017], control [Marti, 2017], etc. In the conventional analysis of stochastic optimization algorithms, it is usually assumed that all data samples are independently and identically distributed (i.i.d.) and queried. For example, data samples in the traditional empirical risk minimization framework are assumed to be queried independently from the underlying data distribution, while data samples in reinforcement learning are assumed to be queried from the stationary distribution of the underlying Markov chain.

Although the i.i.d. data assumption leads to a comprehensive understanding of the statistical limit and computation complexity of SGD, it violates the nature of many practical data-generating stochastic processes, which generate highly correlated samples that depend on the history. In fact, dependent data can be found almost everywhere, e.g., daily stock price [Onalan, 2009, Fort and Roberts, 2005], weather/climate data, state transitions in Markov chains, etc. To understand the impact of data dependence on the convergence and complexity of stochastic algorithms, there is a growing number of recent works that introduce various definitions to quantify data dependence. Specifically, to analyze the finite-time convergence of various stochastic reinforcement learning algorithms, recent studies assume that the dependent samples queried from the Markov decision process satisfy a geometric mixing property [Dalal et al., 2018, Zou et al., 2019, Xu and Gu, 2020, Qu and Wierman, 2020], which requires the underlying Markov chain to be uniformly ergodic or has a finite mixing time [Even-Dar et al., 2003]. On the other hand, to analyze the convergence of stochastic optimization algorithms over dependent data, Karimi et al. [2019] assumed the existence of a solution to the Poisson equation associated with the underlying Markov chain, which is a weaker condition than the uniform ergodic condition [Glynn and Meyn, 1996]. Moreover, Agarwal and Duchi [2012] introduced a $\phi$-mixing process that quantifies how fast the distribution of future data samples (conditioned on a fixed filtration) converges to the underlying stationary data distribution. In particular, the $\phi$-mixing process is more general than the previous two notions of data dependence [Douc et al., 2018].

While the aforementioned works leveraged the above notions of data dependence to characterize the sample complexity of various stochastic algorithms over dependent data,

*Accepted for the 38th Conference on Uncertainty in Artificial Intelligence* (UAI 2022).

there still lacks theoretical understanding of how algorithm structure affects the sample complexity of stochastic algorithms under different levels of data dependence. In particular, a key algorithm structure is the stochastic data sampling scheme, which critically affects the bias and variance of the stochastic learning process. In fact, under i.i.d. data and convex geometry, it is well known that SGD achieves the sample complexity lower bound under various stochastic data sampling schemes [Bottou, 2010], e.g., single-sample sampling and mini-batch sampling. However, these schemes may lead to substantially different convergence behaviors over highly dependent data, as they are no longer unbiased. Therefore, it is of vital importance to understand the interplay among data dependence, stochastic data sampling and sample complexity of stochastic learning algorithms, and we want to ask the following fundamental question.

- **Q:** How does stochastic data sampling affect the convergence rate and sample complexity of stochastic learning algorithms over dependent data?

In this paper, we provide comprehensive answers to this fundamental question. Specifically, we conduct a comprehensive study of the convergence rate and sample complexity of the online SGD algorithm over a wide spectrum of data dependence levels under various stochastic data sampling schemes, including periodic subsampling and mini-batch sampling. Our results show that online SGD with both data sampling schemes achieves a substantially improved sample complexity over the standard online SGD over highly dependent data. We summarize our contributions as follows.

## 1.1 OUR CONTRIBUTIONS

We consider the following stochastic optimization problem.

$$\min_{w \in \mathcal{W}} f(w) := \mathbb{E}_{\xi \sim \mu} \left[ F(w; \xi) \right], \tag{P}$$

where the objective function $f$ is convex and Lipschitz continuous, and the expectation is taken over the stationary distribution $\mu$ of the underlying data-generating process $\mathbf{P}$. To perform online learning, we query a stream of dependent data samples from the underlying data-generating process. Specifically, we adopt the $\phi$-mixing process to quantify the data dependence via a decaying mixing coefficient function $\phi_\xi(k)$ (see Definition 2.2) [Agarwal and Duchi, 2012]. We study the convergence of the online stochastic gradient descent (SGD) algorithm over a $\phi$-mixing data stream under various stochastic data sampling schemes, including periodic subsampling and mini-batch sampling. We summarize and compare the sample complexities of online SGD with these data sampling schemes under different $\phi$-mixing data dependence models in Table 1.

We first study the convergence of online SGD over $\phi$-mixing dependent data samples under the data subsampling scheme.

In particular, the data subsampling scheme utilizes only one data sample per $r$ consecutive data samples by periodically skipping $r-1$ samples. With this data subsampling scheme, the subsampled data samples are less dependent for a larger subsampling period $r$. Also, the improvement is substantial when the data is highly dependent with an algebraic decaying $\phi$-mixing coefficient.

Moreover, we study the sample complexity of online SGD over $\phi$-mixing dependent data samples under the mini-batch sampling scheme. Compare to the data subsampling scheme, mini-batch sampling substantially reduces the mini-batch data dependence without skipping data samples. Consequently, mini-batch update leverages the sample average over a mini batch of data samples to reduce both the bias (caused by the data dependence) and the variance (caused by stochastic sampling). Specifically, we show that online SGD with mini-batch sampling achieves an orderwise lower sample complexity than both the standard online SGD and the online SGD with data subsampling in the full spectrum of the convergence rate of the $\phi$-mixing coefficient. Our study reveals that the widely used mini-batch sampling scheme can effectively reduce the bias caused by data dependence without sacrificing data efficiency.

## 1.2 RELATED WORK

**Stochastic Algorithms over Dependent Data** Steinwart and Christmann [2009] and Modha and Masry [1996] established the convergence analysis of online stochastic algorithms for streaming data with geometric ergodicity. Duchi et al. [2011] proved that the stochastic subgradient method has strong convergence guarantee if the mixing time is uniformly bounded. Agarwal and Duchi [2012] studied the convex/strongly convex stochastic optimization problem and proved high-probability convergence bounds for general stochastic algorithms under general stationary mixing processes. Godichon-Baggioni et al. [2021] provided the non-asymptotic analysis of stochastic algorithms with strongly convex objective function over streaming mini-batch data. In a more general setting, the stochastic approximation (SA) problem was studied in [Karimi et al., 2019] by assuming the existence of solution to a Poisson equation. Recently, Debavelaere et al. [2021] developed the asymptotic convergence analysis of SA problem for sub-geometric Markov dynamic noises.

**Finite-time convergence of reinforcement learning** Recently, a series of work studied the finite-time convergence of many stochastic reinforcement learning algorithms over Markovian dependent samples, including TD learning [Dalal et al., 2018, Xu et al., 2019, Kaledin et al., 2020], Q-learning [Qu and Wierman, 2020, Li et al., 2021, Melo et al., 2008, Chen et al., 2019, Xu and Gu, 2020], fitted Q-iteration [Mnih et al., 2013, 2015, Agarwal et al., 2021],

Table 1: Comparison of sample complexities of SGD, SGD with subsampling and mini-batch sampling under different data dependence models for achieving $f(w) - f(w^*) \leq \epsilon$. Note that $\theta$ parameterizes convergence rate of the $\phi$-mixing coefficient.

| Data dependence model | $\phi_\xi(k)$ | SGD | SGD w/ subsampling | Mini-batch SGD |
|---|---|---|---|---|
| Geometric $\phi$-mixing (Weakly dependent) | $\exp(-k^\theta)$, $\theta > 0$ | $\mathcal{O}(\epsilon^{-2}(\log \epsilon^{-1})^{\frac{2}{\theta}})$ | $\mathcal{O}(\epsilon^{-2}(\log \epsilon^{-1})^{\frac{1}{\theta}})$ | $\mathcal{O}(\epsilon^{-2})$ |
| Fast algebraic $\phi$-mixing (Medium dependent) | $k^{-\theta}$, $\theta \geq 1$ | $\mathcal{O}(\epsilon^{-2-\frac{2}{\theta}})$ | $\mathcal{O}(\epsilon^{-2-\frac{1}{\theta}})$ | $\widetilde{\mathcal{O}}(\epsilon^{-2})$ |
| Slow algebraic $\phi$-mixing (Highly dependent) | $k^{-\theta}$, $0 < \theta < 1$ | $\mathcal{O}(\epsilon^{-2-\frac{2}{\theta}})$ | $\mathcal{O}(\epsilon^{-2-\frac{1}{\theta}})$ | $\mathcal{O}(\epsilon^{-1-\frac{1}{\theta}})$ |

actor-critic algorithms [Wang et al., 2019, Yang et al., 2019, Kumar et al., 2019, Qiu et al., 2019, Wu et al., 2020, Xu et al., 2020], etc. In these studies, the dependent Markovian samples are assumed to be generated from a geometric $\phi$-mixing process, which is satisfied when the underlying Markov chain is uniformly ergodic or time-homogeneous with finite-states.

**Regret of Stochastic Convex Optimization** There have been many known regret bounds for online convex optimization problem. Hazan [2017] has built the standard $\mathcal{O}(\sqrt{T})$ regret bound for online SGD algorithm with assuming the bounded gradient. Xiao [2009] introduces the regret bound of online dual averaging method. To our best knowledge, there is no high-probability guaranteed regret bound for mini-batch SGD with considering the data dependence.

## 2 FORMULATION AND ASSUMPTIONS

In this section, we introduce the problem formulation and some basic assumptions. Consider a model with parameters $w$. For any data sample $\xi$, denote $F(w; \xi) \in \mathbb{R}$ as the sample loss of this data sample under the model $w$. In this paper, we consider the following standard stochastic optimization problem that has broad applications in machine learning.

$$\min_{w \in \mathcal{W}} f(w) := \mathbb{E}_{\xi \sim \mu}\big[ F(w; \xi) \big]. \tag{P}$$

Here, the expectation is taken over the randomness of the data sample $\xi$, which is drawn from an underlying distribution $\mu$. We make the following standard assumptions regarding the problem (P) [Agarwal and Duchi, 2012].

**Assumption 2.1.** *The optimization problem (P) satisfies*

1. *For every $\xi$, function $F(\cdot; \xi)$ is G-Lipschitz continuous over the domain $\mathcal{W}$.*

2. *Function $f(\cdot)$ is convex and bounded below, i.e., $f(w^*) := \inf_{w \in \mathcal{W}} f(w) > -\infty$.*

3. *$\mathcal{W}$ is convex and compact with bounded diameter $R$.*

To solve this stochastic optimization problem, one often needs to query a stream of data samples from the distribution $\mu$ to perform optimization. Unlike traditional stochastic optimization that usually assumes that the data samples are i.i.d. we consider a more general and practical dependent data-generating process as we elaborate below.

**Dependent data-generating process:** We consider a stochastic process $\mathbf{P}$ that generates a stream of data samples $\{\xi_1, \xi_2, ..., \}$, which are not necessarily independent. In particular, the stochastic process $\mathbf{P}$ has an underlying stationary distribution $\mu$. To quantify the dependence of the data generation process, we introduce the following standard $\phi$-mixing process [Agarwal and Duchi, 2012], where we denote $\{\mathcal{F}_t\}_t$ as the filtration generated by $\{\xi_t\}_t$.

**Definition 2.2** ($\phi$-mixing process). Consider a stochastic process $\{\xi_t\}_t$ with a stationary distribution $\mu$. Let $\mathbb{P}(\xi_{t+k} \in \cdot | \mathcal{F}_t)$ be the distribution of the $(t+k)$-th sample conditioned on $\mathcal{F}_t$, and denote $d_{\mathrm{TV}}$ as the total variation distance. Then, the process $\{\xi_t\}_t$ is called $\phi$-mixing if the following mixing coefficient $\phi_\xi(\cdot)$ converges to 0 as $k$ tends to infinity.

$$\phi_\xi(k) := \sup_{t \in \mathbb{N}, A \in \mathcal{F}_t} 2d_{\mathrm{TV}}\big(\mathbb{P}(\xi_{t+k} \in \cdot | A), \mu\big).$$

Intuitively, the $\phi$-mixing coefficient describes how fast the distribution of sample $\xi_{t+k}$ converges to the stationary distribution $\mu$ when conditioned on the filtration $\mathcal{F}_t$, as the time gap $k \to \infty$. The $\phi$-mixing process can be found in many applications, which involve mixing coefficients that converge to zero at different convergence rates. Below we mention some representative examples.

- **Geometric $\phi$-mixing process.** Such a type of process has a geometrically diminishing mixing coefficient, i.e., $\phi_\xi(k) \leq \phi_0 \exp(-ck^\theta)$ for some $\phi_0, c, \theta > 0$. Examples include finite-state ergodic Markov chains and some aperiodic Harris-recurrent Markov processes [Modha and Masry, 1996, Agarwal and Duchi, 2012, Meyn and Tweedie, 2012];

- **Algebraic $\phi$-mixing process.** Such a type of process has a polynomially diminishing mixing coefficient, i.e.,

$\phi_\xi(k) \leq \phi_0 k^{-\theta}$ for some $\phi_0, \theta > 0$. Examples include a large class of Metropolis-Hastings samplers [Jarner and Roberts, 2002] and some queuing systems [Agarwal and Duchi, 2012].

# 3 COMPLEXITY OF ONLINE SGD OVER DEPENDENT DATA

In this section, we recap the convergence results of online SGD over dependent data established in [Agarwal and Duchi, 2012]. Throughout, we define the sample complexity as the total number of samples required for the algorithm to output a model $w$ that achieves an $\epsilon$ convergence error with a certain probability, i.e., $f(w) - f(w^*) \leq \epsilon$ with probability $1 - \delta$. Also, the standard regret of an online learning algorithm is defined as

$$\text{(Regret):} \quad \mathfrak{R}_n := \sum_{t=1}^n F(w(t); \xi_t) - F(w^*; \xi_t),$$

where the models $\{w_1, w_2, ..., w_n\}$ are generated using the data samples $\{\xi_1, \xi_2, ..., \xi_n\}$, respectively, and $w^*$ is the minimizer of $f(w)$. For this sequence of models $\{w_1, w_2, ..., w_n\}$, we make the following mild assumption, which is satisfied by many SGD-type algorithms.

**Assumption 3.1.** *There is a non-increasing sequence $\{\kappa(t)\}_t$ such that $\|w(t+1) - w(t)\| \leq \kappa(t)$.*

Online SGD is a popular and standard algorithm for solving the problem (P). In every iteration $t$, online SGD queries a sample $\xi_t$ from the data-generating process and performs the following SGD update.

$$\text{(SGD):} \quad w(t+1) = w(t) - \eta_t \nabla F(w(t); \xi_t), \quad (1)$$

where $\eta_t$ is the learning rate. In Theorem 2 of [Agarwal and Duchi, 2012], the authors established a high probability convergence error bound for a generic class of stochastic algorithms. Specifically, under the Assumptions 2.1 and 3.1, they showed that for any $\tau \in \mathbb{N}$ with probability at least $1 - \delta$, the averaged predictor $\widehat{w}_n := \frac{1}{n} \sum_{t=1}^n w(t)$ satisfies

$$f(\widehat{w}_n) - f(w^*)$$
$$\leq \frac{\mathfrak{R}_n}{n} + \frac{(\tau-1)G}{n} \sum_{t=1}^n \kappa(t) \quad (2)$$
$$+ \frac{2(\tau-1)GR}{n} + 2GR\sqrt{\frac{2\tau}{n} \log \frac{\tau}{\delta}} + \phi_\xi(\tau)GR.$$

Here, $\mathfrak{R}_n$ is the regret of the algorithm of interest, G is the Lipschitz constant of the loss function $F(\cdot; \xi)$, and $R$ is the diameter of the parameter domain, and $\tau \in \mathbb{N}$ is an auxiliary parameter that is introduced to decouple the dependence of the data samples. From the above bound, one can see that the optimal choice of $\tau$ depends on the convergence rate of the

mixing coefficient $\phi_\xi(\tau)$. Specifically, consider the online SGD algorithm in (1). It can be shown that it achieves the regret $\mathfrak{R}_n = \mathcal{O}(\sqrt{n})$ and satisfies $\kappa(t) = \mathcal{O}(1/\sqrt{t})$ under a proper diminishing learning rate. Consequently, the above high-probability convergence bound for online SGD reduces to

$$f(\widehat{w}_n) - f(w^*)$$
$$\leq \mathcal{O}\Big(\frac{1}{\sqrt{n}} + \inf_{\tau \in \mathbb{N}} \Big\{\frac{\tau-1}{\sqrt{n}} + \sqrt{\frac{\tau}{n} \log \frac{\tau}{\delta}} + \phi_\xi(\tau)\Big\}\Big).$$

Such a bound further implies the following sample complexity results of online SGD under different $\phi$-mixing models.

**Corollary 3.2.** *The sample complexity of online SGD for achieving an $\epsilon$ convergence error over $\phi$-mixing data is*

- *If the data is geometric $\phi$-mixing with parameter $\theta > 0$, then we set $\tau = \mathcal{O}\big((\log \frac{1}{\epsilon})^{\frac{1}{\theta}}\big)$. The resulting sample complexity is in the order of $n = \mathcal{O}\big(\epsilon^{-2}(\log \frac{1}{\epsilon})^{\frac{2}{\theta}}\big)$.*

- *If the data is algebraic $\phi$-mixing with parameter $\theta > 0$, then we set $\tau = \mathcal{O}(\epsilon^{-\frac{1}{\theta}})$. The resulting sample complexity is in the order of $n = \mathcal{O}(\epsilon^{-2-\frac{2}{\theta}})$.*

It can be seen that if the data-generating process has a fast geometrically diminishing mixing coefficient, i.e., the data samples are close to being independent from each other, then the resulting sample complexity is almost the same as that of SGD with i.i.d. samples. On the other hand, if the data-generating process mixes slowly with an algebraically diminishing mixing coefficient, i.e., the data samples are highly dependent, then the data dependence increases the sample complexity by a non-negligible factor of $\epsilon^{-\frac{2}{\theta}}$. In particular, such a factor is substantially large if the mixing rate parameter $\theta$ is close to zero.

# 4 COMPLEXITY OF ONLINE SGD WITH DATA SUBSAMPLING

When apply online SGD to solve stochastic optimization problems over dependent data, the key challenge is that the data dependence introduces non-negligible bias that slows down the convergence of the algorithm. Hence, a straightforward solution is to reduce data dependence before performing stochastic optimization, and data subsampling is such a simple and effective approach [Nagaraj et al., 2020, Kotsalis et al., 2020].

Specifically, consider a stream of $\phi$-mixing data samples $\{\xi_1, \xi_2, \xi_3, \dots\}$. Instead of utilizing the entire stream of data, we subsample a subset of this data stream with period $r \in \mathbb{N}$ and obtain the following subsampled data stream

$$\{\xi_1, \xi_{r+1}, \xi_{2r+1}, \dots\}.$$

In particular, let $\{\mathcal{F}_t\}_t$ be the canonical filtration generated by $\{\xi_{tr+1}\}_t$. Since the consecutive subsampled samples are

$r$ time steps away from each other, it is easy to verify that the subsampled data stream $\{\xi_{tr+1}\}_t$ is also a $\phi$-mixing process with mixing coefficient given by $\phi_\xi^r(t) = \phi_\xi(rt)$, where $\phi_\xi^r$ denotes the mixing coefficient of the subsampled data stream $\{\xi_{tr+1}\}_t$. Therefore, by periodically subsampling the data stream, the resulting subsampled process has a faster-converging mixing coefficient. Then, we can apply online SGD to such subsampled data, i.e.,

(SGD with subsampling):
$$w(t+1) = w(t) - \eta_t \nabla F(w(t); \xi_{tr+1}). \quad (3)$$

In particular, the convergence error bound in eq. (2) still holds by replacing $\phi_\xi(\tau)$ with $\phi_\xi(r\tau)$, and we obtain the following bound for online SGD with subsampling.

$$f(\widehat{w}_n) - f(w^*) \quad (4)$$
$$\leq \mathcal{O}\Big(\frac{1}{\sqrt{n}} + \inf_{\tau \in \mathbb{N}} \Big\{\frac{(\tau-1)}{\sqrt{n}} + \sqrt{\frac{\tau}{n} \log \frac{\tau}{\delta}} + \phi_\xi(r\tau)\Big\}\Big).$$

Such a bound implies the following sample complexity results of online SGD with subsampling under different convergence rates of the mixing coefficient $\phi_\xi$.

**Corollary 4.1.** *The sample complexity of online SGD with subsampling for achieving an $\epsilon$ convergence error over $\phi$-mixing data process is.*

- *If the data is geometric $\phi$-mixing with parameter $\theta > 0$, then we choose $r = \mathcal{O}\big((\log \frac{1}{\epsilon})^{\frac{1}{\theta}}\big)$ and $\tau = \mathcal{O}(1)$. The resulting sample complexity is $rn = \mathcal{O}\big(\epsilon^{-2}(\log \frac{1}{\epsilon})^{\frac{1}{\theta}}\big)$.*

- *If the data is algebraic $\phi$-mixing with parameter $\theta > 0$, then we choose $r = \mathcal{O}\big(\epsilon^{-\frac{1}{\theta}}\big)$ and $\tau = \mathcal{O}(1)$. The resulting sample complexity is $rn = \mathcal{O}\big(\epsilon^{-2-\frac{1}{\theta}}\big)$.*

Compare the above sample complexity results with those of the standard online SGD in Corollary 3.2, we conclude that data-subsampling can improve the sample complexity by a factor of $(\log \frac{1}{\epsilon})^{\frac{1}{\theta}}$ and $\epsilon^{-\frac{1}{\theta}}$ for geometric $\phi$-mixing and algebraic $\phi$-mixing data process, respectively. Intuitively, this is because with data subsampling, we can choose a sufficiently large subsampling period $r$ to decouple the data dependence in the term $\phi_\xi(r\tau)$, as opposed to choosing a large $\tau$ in Corollary 3.2. In this way, the order of the dominant term $\sqrt{\frac{\tau}{n} \log \frac{\tau}{\delta}}$ is reduced. Therefore, when the data is highly dependent, it is beneficial to subsample the dependent data before performing SGD. We also note another advantage of using data-subsampling, i.e., it only requires computing the stochastic gradients of the subsampled data, and therefore can substantially reduce the computation complexity.

## 5 COMPLEXITY OF ONLINE SGD WITH MINI-BATCH SAMPLING

Although the data-subsampling scheme studied in the previous section helps improve the sample complexity of online SGD, it does not leverage the full information of all the queried data. In particular, when the data is highly dependent, we need to choose a large period $r$ to reduce data dependence, and this will throw away a huge amount of valuable samples. In this section, we study online SGD with another popular data sampling scheme that leverages the full information of all the sampled data, i.e., the mini-batch sampling scheme. We show that this simple and widely used scheme can effectively reduce data dependence without skipping data samples, and can achieve an improved sample complexity over online SGD with subsampling.

Specifically, consider a data stream $\{\xi_t\}_t$ with $\phi$-mixing dependent samples. We rearrange the data samples into a stream of mini-batches $\{x_t\}_t$, where each mini-batch $x_t$ contains $B$ samples, i.e., $x_t = \{\xi_{(t-1)B+1}, \xi_{(t-1)B+2}, \ldots, \xi_{tB}\}$. Then, we perform mini-batch SGD update as follows.

(SGD with mini-batch sampling):
$$w(t+1) = w(t) - \frac{\eta_t}{B} \sum_{\xi \in x_t} \nabla F(w(t); \xi). \quad (5)$$

Performing online learning with mini-batch sampling has several advantages. First, it substantially reduce the optimization variance and allows to use a large learning rate to facilitate the convergence of the algorithm. As a comparison, SGD with subsampling suffers from a large optimization variance. Second, unlike subsampling, mini-batch sampling utilizes the information of all the queried data samples to improve the performance of the model. Moreover, as we show in the following lemma, mini-batch sampling substantially reduces the stochastic bias caused by the data dependence. In the sequel, we denote $F(w; x) := \frac{1}{B} \sum_{\xi \in x} F(w; \xi)$ as the average loss on a mini-batch of samples. With a bit abuse of notation, we also define $\{\mathcal{F}_t\}_t$ as the canonical filtration generated by the mini-batch samples $\{x_t\}_t$.

**Lemma 5.1.** *Let Assumption 2.1 hold and consider the mini-batch data stream $\{x_t\}_t$. Then, for any $w, v \in \mathcal{W}$ measureable with regard to $\mathcal{F}_t$ and any $\tau \in \mathbb{N}$, it holds that*

$$\mathbb{E}\big[F(w; x_{t+\tau}) - F(v; x_{t+\tau})|\mathcal{F}_t\big] - \big(f(w) - f(v)\big)$$
$$\leq \frac{GR}{B} \sum_{i=1}^{B} \phi_\xi(\tau B + i). \quad (6)$$

With dependent data, the above lemma shows that we can approximate the population risk $f(w)$ by the conditional expectation $\mathbb{E}[F(w; x_{t+\tau})|\mathcal{F}_t]$, which involves the mini-batch $x_{t+\tau}$ that is $\tau$ steps ahead of the filtration $\mathcal{F}_t$. Intuitively, by the definition of $\phi$-mixing process, as $\tau$ gets larger, the distribution of $x_{t+\tau}$ conditional on $\mathcal{F}_t$ gets closer to the stationary distribution $\mu$. In general, the estimation bias $\frac{GR}{B} \sum_{i=1}^{B} \phi_\xi(\tau B + i)$ depends on both the batch size and the accumulated mixing coefficient over the corresponding

batch of samples. To provide a concrete understanding, below we calculate the estimation bias in eq. (6) for various $\phi$-mixing processes.

- **Geometric $\phi$-mixing:** In this case, $\sum_{i=1}^{B} \phi_\xi(\tau B + i) \leq \sum_{i=1}^{\infty} \phi_\xi(i) = \mathcal{O}(1)$. Hence, the estimation bias is in the order of $\mathcal{O}(\frac{GR}{B})$.

- **Fast algebraic $\phi$-mixing ($\theta \geq 1$):** In this case, $\sum_{i=1}^{B} \phi_\xi(\tau B + i) \leq \sum_{i=1}^{\infty} \phi_\xi(i) = \widetilde{\mathcal{O}}(1)$. Hence, the estimation bias is in the order of $\widetilde{\mathcal{O}}(\frac{GR}{B})$, where $\widetilde{\mathcal{O}}$ hides all logarithm factors.

- **Slow algebraic $\phi$-mixing ($0 < \theta < 1$):** In this case, $\sum_{i=1}^{B} \phi_\xi(\tau B + i) \leq \mathcal{O}((\tau B)^{1-\theta})$. Hence, the estimation bias is in the order of $\mathcal{O}(\frac{GR\tau^{1-\theta}}{B^\theta})$.

It can be seen that if the mixing coefficient converges fast, i.e., either geometrically or fast algebraically, then the data dependence has a negligible impact on the estimation error. On the other hand, when the mixing coefficient converges slow algebraically, it substantially increases the estimation bias, but it is still beneficial to use a large batch size.

We obtain the following convergence error bound for online SGD with mini-batch sampling over dependent data.

**Theorem 5.2.** *Let Assumption 2.1 and 3.1 hold. Apply SGD with mini-batch sampling to solve the stochastic optimization problem (P) over $\phi$-mixing dependent data process and assume that it achieves regret $\mathfrak{R}_n$. Then, for any $\tau \in \mathbb{N}$ and any minimizer $w^*$ with probability at least $1 - \delta$, the averaged predictor $\widehat{w}_n := \frac{1}{n}\sum_{t=1}^{n} w(t)$ satisfies*

$$f(\widehat{w}_n) - f(w^*)$$
$$\leq \frac{\mathfrak{R}_n}{n} + \frac{G(\tau - 1)}{n}\sum_{t=1}^{n-\tau+1} \kappa(t) + \frac{2GR(\tau - 1)}{n}$$
$$+ \mathcal{O}\left(\frac{1}{nB}\sum_{i=1}^{B} \phi(\tau B + i)\right.$$
$$\left. + \sqrt{\frac{\tau}{nB}\log\frac{\tau}{\delta}}\log\frac{n}{\delta}\left(B^{-\frac{1}{4}} + \left[\sum_{i=1}^{B} \phi(i)\right]^{\frac{1}{4}}\right)\right). \quad (7)$$

To further understand the order of the above bound, a standard regret analysis shows that mini-batch SGD achieves the regret $\frac{\mathfrak{R}_n}{n} = \widetilde{\mathcal{O}}(\sqrt{\frac{\sum_{j=1}^{n} \phi(j)}{nB}})$ and $\kappa(t) \equiv \mathcal{O}(\sqrt{\frac{B}{n}})$ (see Theorem C.3 for the proof). Consequently, the above convergence error bound reduces to the following bound.

$$f(\widehat{w}_n) - f(w^*)$$
$$\leq \widetilde{\mathcal{O}}\left(\sqrt{\frac{\sum_{j=1}^{n} \phi(j)}{nB}} + \frac{GR(\tau - 1)}{n}\right.$$
$$\left. + \frac{1}{nB}\sum_{i=1}^{B} \phi(\tau B + i) + \sqrt{\frac{\tau}{nB}}\left(B^{-\frac{1}{4}} + \left[\sum_{i=1}^{B} \phi(i)\right]^{\frac{1}{4}}\right)\right).$$

Such a bound further implies the following sample complexity results of online SGD with mini-batch sampling under different convergence rates of the mixing coefficient $\phi_\xi$.

**Corollary 5.3.** *The sample complexity of online SGD with mini-batch sampling for achieving an $\epsilon$ convergence error over $\phi$-mixing dependent data is*

- *If the data is geometric $\phi$-mixing with parameter $\theta > 0$, then we choose $\tau = 1, B = \mathcal{O}(\epsilon^{-1}), n = \mathcal{O}(\epsilon^{-1})$. The overall sample complexity is $nB = \mathcal{O}(\epsilon^{-2})$.*

- *If the data is fast algebraic $\phi$-mixing with parameter $\theta \geq 1$, then we choose $\tau = 1, B = \mathcal{O}(\epsilon^{-1}), n = \mathcal{O}(\epsilon^{-1})$. The overall sample complexity is $nB = \widetilde{\mathcal{O}}(\epsilon^{-2})$.*

- *If the data is slow algebraic $\phi$-mixing with parameter $0 < \theta < 1$, then we choose $\tau = 1, B = \mathcal{O}(\epsilon^{-\frac{1}{\theta}}), n = \mathcal{O}(\epsilon^{-1})$. The overall sample complexity is $nB = \mathcal{O}(\epsilon^{-1-\frac{1}{\theta}})$.*

*Remark.* This corollary provides a potential way to set the optimal batch size $B$ with respect to the mixing rate $\theta$. Specifically, we can leverage Lemma 5.1 to estimate the dependence parameter $\theta$. Choosing batch size $B = 1$, the upper bound of Lemma 5.1 becomes $GR\phi_\xi(\tau + 1)$, which is proportional to the mixing coefficient $\phi_\xi(\tau + 1)$. Therefore, the left-hand side $\mathbb{E}\big[F(w; x_{t+\tau}) - F(v; x_{t+\tau})|\mathcal{F}_t\big] - \big(f(w) - f(v)\big)$ of Lemma 5.1 serves as an estimator, which can be estimated by (conditional) sample average queried at any fixed points $\omega, v$. Once we estimate this quantity with various values of $\tau$, we can use regression to find out the type of convergence for $\phi_\xi(\tau)$ and estimate the parameter $\theta$. With the estimated $\theta$, we then follow this corollary to choose the batch size.

It can be seen that online SGD with mini-batch sampling improves the sample complexity of online SGD with sub-sampling by a factor of $\mathcal{O}((\log\frac{1}{\epsilon})^{\frac{1}{\theta}})$, $\widetilde{\mathcal{O}}(\epsilon^{-\frac{1}{\theta}})$ and $\mathcal{O}(\epsilon^{-1})$ for geometric $\phi$-mixing, fast algebraic $\phi$-mixing and slow algebraic $\phi$-mixing data samples, respectively. This shows that mini-batch sampling can effectively reduce the bias caused by data dependence and leverage the full information of all the data samples to improve the learning performance.

To provide an intuitive explanation, this is because with mini-batch sampling, we can choose a sufficiently large batch size $B$ to reduce the bias caused by the data dependence and then choose a small auxiliary parameter $\tau = 1$. As a comparison, to control the bias caused by data dependence, the standard online SGD needs to choose a very large $\tau$ and the online SGD with subsampling needs to choose a large subsampling period $r$ that skips a huge amount of valuable data samples, especially when the mixing coefficient converges slowly. Therefore, our result proves that it is beneficial to use mini-batch data sampling when the data samples are highly dependent.

Our proof of the high-probability bound in Theorem 5.2 for SGD with mini-batch sampling involves substantial new developments compared with the proof of [Agarwal and Duchi, 2012]. Next, we elaborate on our technical novelty.

- In [Agarwal and Duchi, 2012], they defined the following random variable

$$X_t^i := f\big(w((t-1)\tau + i)\big) - f(w^*)$$
$$+ F\big(w((t-1)\tau + i); \xi_{t+\tau-1}\big) - F\big(w^*; \xi_{t+\tau-1}\big).$$

As this random variable involves only one sample $\xi_{t+\tau-1}$, they bound the bias term $X_t^i - \mathbb{E}[X_t^i | \mathcal{F}_{t-1}]$ as a universal constant. As a comparison, the random variable $X_t^i$ would involve a mini-batch of samples $x_{t+\tau-1}$ in our analysis. With the mini-batch structure, the bias $X_t^i - \mathbb{E}[X_t^i | \mathcal{F}_{t-1}^i]$ can be written as an average of $B$ zero-mean dependent random variables, which is close to zero with high probability due to the concentration phenomenon. Consequently, we are able to apply a Bernstein-type inequality developed in [Delyon et al., 2009] for dependent stochastic process to obtain an improved bias bound from $\mathcal{O}(1)$ to $\widetilde{\mathcal{O}}(1/\sqrt{B})$. This is critical for obtaining the improved bound.

- Second, with the improved high-probability bias bound mentioned above, the remaining proof of [Agarwal and Duchi, 2012] no longer holds. Specifically, we can no longer apply the Azuma's inequality to bound the accumulated bias $\sum_t (X_t^i - \mathbb{E}[X_t^i | \mathcal{F}_{t-1}^i])$, as each bias term is no longer bounded with probability one. To address this issue, we developed a generalized Azuma's inequality for martingale differences in Lemma B.3 based on Proposition 34 of [Tao et al., 2015] for independent zero-mean random variables.

- Third, we develop a high-probability regret bound for online SGD with mini-batch sampling over dependent data so that it can be integrated with the high-probability convergence bound in Theorem 5.2. To our best knowledge, the regret of SGD over dependent data has not been studied before.

# 6 EXPERIMENTS

In this section, we examine our SGD theory via two experiments on stochastic quadratic programming and neural network training with dependent data.

## 6.1 STOCHASTIC QUADRATIC PROGRAMMING

We consider the following stochastic convex quadratic optimization problem.

$$\min_{w \in \mathbb{R}^d} f(w) := \mathbb{E}_{\xi \sim \mu}\big[(w - \xi)^\top A(w - \xi)\big],$$

where $A \succeq 0$ is a fixed positive semi-definite matrix and $\mu$ is the uniform distribution on $[0, 1]^d$. Then, following the construction in [Jarner and Roberts, 2002], we generate an algebraic $\phi$-mixing Markov chain that has the stationary distribution $\mu$. In particular, its mixing coefficient $\phi_\xi(k)$ converges at a sublinear convergence rate $k^{-\frac{1}{r}}$, where $r > 0$ is a parameter that controls the speed of convergence. Please refer to Appendix D for more details of the experiment setup.

We first estimate the following stochastic bias at the fixed origin point $w = \mathbf{0}_d$.

$$(\text{Bias}): \quad \Big|\mathbb{E}\big[F(w; x_\tau) | x_0 = \mathbf{0}_d\big] - f(w)\Big|,$$

where the expectation is taken over the randomness of the mini-batch of samples queried at time $\tau \in \mathbb{N}$. Such a bias is affected by several factors, including the time gap $\tau$, the batch size $B$ and the convergence rate parameter $r$ of the mixing coefficient.

In Figure 1, we investigate the impact of these factors on the stochastic bias, and we use 10k Monte Carlo samples to estimate the stochastic bias. The top two figures fix the batch size, and it can be seen that the bias decreases as $\tau$ increases, which matches the definition of the $\phi$-mixing process. Also, a faster-mixing Markov chain (i.e., smaller $r$) leads to a smaller bias. In particular, with batch size $B = 1$ and a slow-mixing chain $r = 2$, it takes an unacceptably large $\tau$ to achieve a relatively small bias. This provides an empirical justification to Corollary 3.2 and explains why the standard SGD suffers from a high sample complexity over highly dependent data. Moreover, as the batch size gets larger, one can achieve a numerically smaller bias, which matches our Lemma 5.1. The bottom two figures fix the convergence rate parameter of the mixing coefficient, and it can be seen that increasing the batch size significantly reduces the bias. Consequently, instead of choosing a large $\tau$ to reduce the bias, one can simply choose a large batch size $B = 100$ and set $\tau = 1$. This observation matches and justifies our theoretical results in Corollary 5.3.

We further compare the convergence of SGD, SGD with subsampling and mini-batch SGD. Here, we set $r = 2$ to generate highly dependent data samples. We set learning rate $\eta = 0.01$ for both SGD and SGD with subsampling, and set learning rate $\eta = 0.01 \times \sqrt{\frac{B}{\sum_{j=1}^{B} \phi_\xi(j)}} = 0.01 \times 100^{1/4}$ for mini-batch SGD with batch size $B = 100$, as suggested by Theorem C.3 in the appendix. The results are plotted in Figure 2, where each curve corresponds to the mean of 100 independent trails. It can be seen that SGD with subsampling achieves a lower loss than the standard SGD asymptotically, due to the use of less dependent data. Moreover, mini-batch SGD achieves the smallest asymptotic loss. All these observations are consistent with our results.

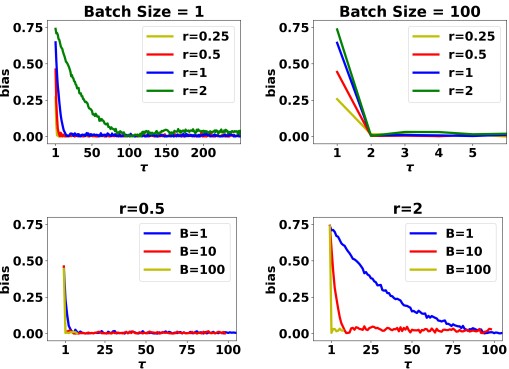

Figure 1: Impact of $\tau$, batch size $B$ and convergence rate of mixing coefficient on the bias in quadratic programming.

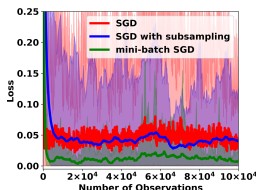

Figure 2: Comparison of sample complexity of different SGD algorithms in quadratic programming.

## 6.2 NEURAL NETWORK TRAINING

We further apply these online SGD algorithms to train a convolutional neural network with the MNIST dataset [Lecun et al., 1998]. The network consists of two convolution blocks followed by two fully connected layers. Specifically, each convolution block contains a convolution layer, a maxpooling layer with stride step 2, and a ReLU activation layer. The convolution layers in the two blocks have input channel 1, 10 and output channel 10, 20, respectively, and both of them have kernel size 5, stride step 1 and with no padding. The two fully connected layers have input dimensions 320, 50 and output dimensions 50, 10, respectively.

To generate a stream of dependent data, we first generate an algebraic $\phi$-mixing Markov chain $\{X_t\}_t$ with the construction provided in [Jarner and Roberts, 2002]. Then, we map each $X_t$ to a label of the MNIST dataset $\{0, 1, 2, \ldots, 9\}$, and uniformly sample an image at random from the corresponding image class. This data-generating process generates a dependent data stream with a $\phi_\xi$-mixing coefficient approximately $k^{-\frac{1}{r}}$.

We first test the performance of SGD with a fixed batch size and different correlation coefficients. Specifically, we choose batch size $B = 1000$ and consider different correlation coefficients $r \in \{1.0, 1.25, 1.5, 1.75, 2.0\}$. Here, a larger $r$ implies higher data dependency. Figure 3 (left) plots the experiment results. It can be seen that with an increasing correlation coefficient, the convergence of SGD is slower.

We further fix the correlation coefficient $r = 2.0$ and vary the batch size $B \in \{8, 16, 32, 64, 128\}$. Figure 3 (right) plots the experiment results. It can be seen that SGD with the largest batch size $B = 128$ achieves the smallest asymptotic loss among all choices of batch sizes. In particular, SGD with a larger batch size tends to converge faster over such dependent data. This also matches our theoretical analysis and it implies that mini-batch SGD with a large batch size can benefit neural network training over dependent data.

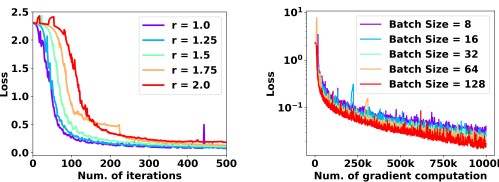

Figure 3: Comparison of SGD over dependent data with different mixing coefficients and batch sizes.

## 7 CONCLUSION

In this study, we investigate the convergence property of SGD under various popular stochastic update schemes over highly dependent data. Unlike the conventional i.i.d. data setting in which the stochastic update schemes do not affect the sample complexity of SGD, the convergence of SGD in the data-dependent setting critically depends on the structure of the stochastic update scheme. In particular, we show that both data subsampling and mini-batch sampling can substantially improve the sample complexity of SGD over highly dependent data. Our study takes one step forward toward understanding the theoretical limits of stochastic optimization over dependent data, and it opens many directions for future study. For example, it is interesting to further explore the impact of algorithm structure on the sample complexity of stochastic reinforcement learning algorithms. Also, it is important to develop advanced algorithm update schemes that can facilitate the convergence of learning over highly dependent data.

### Acknowledgements

The work of Shaocong Ma, Ziyi Chen and Yi Zhou was supported in part by U.S. National Science Foundation under the Grants CCF-2106216 and DMS-2134223.

The work of Y. Liang was supported in part by U.S. National Science Foundation under the grants CCF-1909291 and ECCS-2113860.

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
