# OpenReview forum: "Data Sampling Affects the Complexity of Online SGD over Dependent Data"
_auai.org/UAI/2022/Conference — UAI 2022 Poster_

### Official Review · Reviewer_Hto2 · 2022-04-11

**Q2(1) Originality/Novelty:** 3
**Q2(2) Significance/Impact:** 2
**Q2(3) Correctness/Technical Quality:** 3
**Q2(6) Clarity Of Writing:** 4
**Q6 Overall Score:** 5
**Q8 Confidence In Your Score:** 3

**Q1 Summary And Contributions:**

This paper studies the sample complexity required for online SGD to
obtain $\epsilon$ error when the data is sampled from a $\phi$-mixing stochastic process (not i.i.d.). It establishes rates for
two alternatives to SGD: periodic subsampling and the minibatch
SGD.

The main difference to prior work (specifically, Agrawal & Duchi
(2012)) is the analysis of these two particular modifications to SGD
detailing how the design choice of data sampling affects the
convergence rate/sample complexity.

**Q2 Assessment Of The Paper:**

More detailed information regarding each of these aspects is given below:

**Q2(4) Quality Of Experiments (Optional):**

2: Fair: The experimental evaluation is weak: important baselines are missing, or the results do not adequately support the main claims.

**Q2(5) Reproducibility:**

3: Good: Key resources (e.g., proofs, code, data) are available and key details (e.g., proofs, experimental setup) are sufficiently well-described for competent researchers to confidently reproduce the main results.

**Q3 Main Strengths:**

- Very clear writing and easy-to-follow presentation, despite the fairly involved mathematical content.
- Clear differentiation from prior work in the main text.
- Solid and well-explained technical results.

**Q4 Main Weakness:**

- Missing a convincing example where we only have access to data
  points sampled from a $\phi$-mixing process rather than i.i.d.

- The optimal setting for both the subsampling rate $r$ and (to a
  lesser extent) the batch size $B$ depend on parameter $\theta$ of
  the mixing coefficient. Practical estimation of $\theta$ (or even
  estimating which mixing regime we're in) is not discussed.

- The experiments are unnatural. Rather than using a mixing
  distribution that arises automatically (e.g., if the data were one
  of the types discussed in the intro), the authors synthetically
  generate an algebraically mixing Markov chain and sample the points
  from that.


**Q5 Detailed Comments To The Authors:**

- [Missing convincing example] Some
  hypothetical ones are discussed in the introduction (stock price
  data, weather/climate data) and similar assumptions have been made
  in related areas (analysis for stochastic reinforcement learning
  algorithms). However, I think the question is of theoretical
  interest (what modifications can we make to SGD to make it perform
  well with dependent data?)  and the results and presentation are
  solid, so I don't consider this a big weakness.

- [optimal setting depends on $\theta$] It would
  be interesting to have a short discussion on how to set $B$ or $r$
  in practice using estimations of the mixing, or on the possibility
  of adaptive results that don't need details of $\phi$ to be
  specified.

- the re-labeled data stream notation defined at the beginning of
  section B.2 is used earlier in the proof of Lemma B.1.

- Proposition B.2 seems very similar to Agrawal & Duchi
  proposition 2. Can you comment more on the differences?  It seems
  like A&D have a factor of 2 on the $2\tau G R$ term that is missing
  from your $GR(\tau - 1)$. Should your bound also have the factor of
  2? It seems like from the bound on page 13 that you're saying
  $G(\tau-1)|w(t) - w^*| + G(\tau-1) |w(t) - w^*| \le GR (\tau -1)$,
but shouldn't there be a 2? Does this change the results?

- My proof-checking process was glorified pattern-matching with
  Agrawal & Duchi, but in general it seems like there's significant
  overlap on the proof techniques. More detailed differentiation in
  the appendix would be helpful. I.e., a more detailed version of the
  end of Section 5, in-line with the proofs. The technical novelty
  seems lower after comparing to Agrawal & Duchi line-by-line, so
  further emphasis here would help.

- A proof sketch / summary of the main techniques and challenges
  (compared to Agrawal & Duchi) at the beginning of appendix B would
  be very helpful. As-is, the results in the appendix do not flow naturally and it's hard to get a picture of what's going on.



**Q7 Justification For Your Score:**

I found the writing to be clearly presented and the main results
summarized well.  However, the experiments are unnatural, and
the proofs difficult to follow without a sketch, and none of the main
technical contributions appear in the main text. I would like to see
at least one key technical novelty explained in detail in the main text with a
proof sketch.


**Q9 Complying With Reviewing Instructions:**

1: Yes.

---

### Official Review · Reviewer_2UQP · 2022-04-15

**Q2(1) Originality/Novelty:** 3
**Q2(2) Significance/Impact:** 3
**Q2(3) Correctness/Technical Quality:** 3
**Q2(6) Clarity Of Writing:** 2
**Q6 Overall Score:** 7
**Q8 Confidence In Your Score:** 4

**Q1 Summary And Contributions:**

This paper studies the relation between the data-sampling process and the convergence rate and sample complexity of the online stochastic gradient descent (SGD) algorithm. Specifically, This paper tries to answer the following question.

Question: How does stochastic data sampling affect the convergence rate and sample complexity of stochastic learning algorithms over dependent data?

**Q2 Assessment Of The Paper:**

More detailed information regarding each of these aspects is given below:

**Q2(4) Quality Of Experiments (Optional):**

3: Good: The experimental evaluation is adequate, and the results convincingly support the main claims.

**Q2(5) Reproducibility:**

3: Good: Key resources (e.g., proofs, code, data) are available and key details (e.g., proofs, experimental setup) are sufficiently well-described for competent researchers to confidently reproduce the main results.

**Q3 Main Strengths:**

The results and idea are main strengths of the paper.

**Q4 Main Weakness:**

The most important problem with this paper is its writing and condensed form.
The second problem is that they do not consider the cost of gathering the data when computing the sample complexity.
The third problem is the concluding remarks.

**Q5 Detailed Comments To The Authors:**


Although this paper considers an important and challenging problem, it is not well-written. It is not self-contained, and it is hard to read this paper because the reader should refer to the references and find the meaning of some terms. The main issue with this paper is that when the data is subsampled, we do not use some valuable gathered data while the gathering cost has been paid. This problem becomes more serious when the data is highly dependent, they then should choose a large period r to reduce data dependence, and this will throw away a huge amount of valuable samples. This issue was addressed in section 5.

What do you mean by "\phi-mixing model", "\phi-mixing property", "\phi-mixing process"? Are they the same?

Section 1.1 can be shortened because some parts are redundant and repeated from the introduction, and some parts are also repeated in the next sections.

The definition of sample complexity given in the first paragraph of section 3 is not accurate.

Some equations are labeled with numbers, some others are labeled with some characters and some others are labeled with both. Please unify the numbering.

How do you show that the consecutive subsampled samples are a \phi-mixing process?

The authors stated that the bound given in equation (2) holds also for the consecutive subsampled samples by replacing "\phi_{\xi}(\tau)" with "\phi_{\xi}(r\tau)". The resulting bound is given in equation (4). However, the bound given in equations (4) and (2) have several changes.

The proof of Corollary 4.1 is unclear. Also, I can not obtain such sample complexities from the bound given in equation (4). In addition, the paragraph following Corollary 4.1 is not clear. For instance, the improvement of sample complexity is not clear.

Some notations in equations (2) and (6) are not defined.

When B is too large, the computational complexity of the SGD will be high. How do you overcome this issue?

I did not check the proofs completely. But I checked some of them, and they are correct.

**Q7 Justification For Your Score:**

The introductory parts are too long and the results are presented in condensed form.

The results and idea are main strengths of the paper.

-------------------------------------------------------------

I have read all reviews and the response of the authors.

I think the paper is good but is not solid.

Based on the above discussion my score is between Weak Accept and Accept.


**Q9 Complying With Reviewing Instructions:**

1: Yes.

---

### Official Review · Reviewer_exPY · 2022-04-24

**Q2(1) Originality/Novelty:** 2
**Q2(2) Significance/Impact:** 2
**Q2(3) Correctness/Technical Quality:** 3
**Q2(6) Clarity Of Writing:** 4
**Q6 Overall Score:** 6
**Q8 Confidence In Your Score:** 2

**Q1 Summary And Contributions:**

This paper studies the theoretical sample complexity and convergence rate of online SGD over dependent data. In particular, this work explores the setting of stochastic data sampling schemes, such as data subsampling and mini batch sampling. They show that under certain conditions, data subsampling and mini batch can yield improved bounds for sample complexity. Finally, numerical experiments are conducted over both a synthetic and MNIST experiment to illustrate the theoretical results.

**Q2 Assessment Of The Paper:**

More detailed information regarding each of these aspects is given below:

**Q2(4) Quality Of Experiments (Optional):**

3: Good: The experimental evaluation is adequate, and the results convincingly support the main claims.

**Q2(5) Reproducibility:**

3: Good: Key resources (e.g., proofs, code, data) are available and key details (e.g., proofs, experimental setup) are sufficiently well-described for competent researchers to confidently reproduce the main results.

**Q3 Main Strengths:**

- The paper is well-written and states any assumptions clearly. In addition, previous works are discussed and the contributions of this work are clear.
- The theoretical contributions to the sample complexity of minibatch SGD seems to be novel and confirms some key intuitions, such as the fact that “mini-batch sampling can effectively reduce the bias caused by data dependence”. In addition, the proof to Theorem 5.2 seems to be non-trivial in comparison to existing results for online SGD.

**Q4 Main Weakness:**

- As mentioned in section 6.2, their “theoretical analysis in convex optimisation may also be applicable to nonconvex functions in deep learning” and the main results in the paper assumes convexity for $f$. Therefore the results of 6.2 cannot entirely be interpreted using the developed theoretical bounds, although if this paper also extended the theoretical results to nonconvex settings, such as neural network training.
- It would also have been more interesting to look a few more examples in convex optimisation in the theoretical and experiments section, such as SVMs or boosting algorithms, as suggested in Agarwal and Duchi (2012).
- One of the 2 main results seem to be incremental with respect to Agarwal and Duchi (2012: The proof the results in Section 4 in appendix A seems like a straightforward application of Theorem 2 of Agarwal and Duchi (2012).

**Q5 Detailed Comments To The Authors:**

- There were instances in the paper where $\min_{w\in\mathbb{R}^d}$ was not written on the right hand side of the equation
- Some of the claims were repeated too many times and it would have been better to be more concise e.g. "therefore, mini-batch sampling can effectively reduce the statistical bias of stochastic approximation for a wide spectrum of dependent data generating processes" was repeated 4 times in section 5.

**Q7 Justification For Your Score:**

The paper is well written and the theoretical results presented are rigorous. As the main contribution to the paper, the authors theoretically confirmed, via a non-trivial proof, some intuitions regarding mini-batch sampling for SGD and how it improves the convergence for convex cases. However, it would have been better, just as in Agarwal and Duchi (2012), to illustrate a few more examples from convex optimisation problems. In addition, this work cannot be directly applied to nonconvex problems

**Q9 Complying With Reviewing Instructions:**

1: Yes.

---

### Decision · Program_Chairs · 2022-05-15

**Decision:**

Accept (Poster)

**Comment:**

Meta Review: All reviewers appreciate the clear presentation of the paper and the contribution regarding the theory and the insights. The responses from the author have addressed most of the questions(many regarding experiments )from the reviewers. I recommend this paper be accepted.